# A Novel UHPLC-MS/MS Based Method for Isomeric Separation and Quantitative Determination of Cyanogenic Glycosides in American Elderberry

**DOI:** 10.3390/metabo14070360

**Published:** 2024-06-26

**Authors:** Deepak M. Kasote, Zhentian Lei, Clayton D. Kranawetter, Ashley Conway-Anderson, Barbara W. Sumner, Lloyd W. Sumner

**Affiliations:** 1Metabolomics Center, University of Missouri-Columbia, Columbia, MO 65211, USA; deepakkasote06@gmail.com (D.M.K.); sumnerb@missouri.edu (B.W.S.); 2Department of Biochemistry, University of Missouri-Columbia, Columbia, MO 65211, USA; cdk374@mail.missouri.edu; 3Center for Agroforestry, University of Missouri-Columbia, Columbia, MO 65211, USA; acconway@missouri.edu

**Keywords:** cyanogenic glycosides, elderberry, mobile phase additives, isomeric separation, (*R*)-prunasin, (*S*)-prunasin, sambunigrin, UHPLC-MS/MS

## Abstract

LC-MS/MS analyses have been reported as challenging for the reliable separation and quantification of cyanogenic glycosides (CNGs), especially (*R*)-prunasin and sambunigrin isomers found in American elderberry (*Sambucus nigra* L. subsp. *canadensis* (L.) Bolli). Hence, a novel multiple reaction monitoring (MRM)-based ultra-high-performance liquid chromatography–tandem mass spectrometry (UHPLC-MS/MS) method was developed and validated in the present study for simultaneous separation and quantification of five CNGs, including amygdalin, dhurrin, linamarin, (*R*)-prunasin, and (*S*)-prunasin (commonly referred to as sambunigrin). Initially, the role of ammonium formate was investigated as an aqueous mobile-phase additive in developing MRM-based UHPLC-MS/MS. Later, chromatographic conditions for the resolved separation of (*R*)-prunasin and sambunigrin were identified. Validation studies confirmed that the developed method has good linearity and acceptable precision and accuracy. A noticeable matrix effect (mainly signal enhancement) was observed in leaf samples only. This method was used to detect and quantify CNGs, including (*R*)-prunasin and sambunigrin, in leaf and fruit samples of American elderberry. Among the studied CNGs, only (*R*)-prunasin was detected in the leaf samples. Interestingly, (*S*)-prunasin (sambunigrin) was not detected in the samples analyzed, even though it has been previously reported in elderberry species.

## 1. Introduction

Cyanogenic glycosides (CNGs) are synthesized in many plant species as chemical defense compounds against herbivores and pathogens [1]. CNGs also have a role in nitrogen transport and related primary metabolism of plants [2]. Approximately 120 CNGs have been reported in over 2500 plant species, including food crops such as sorghum, almond, cassava, butter bean, nut, peach, cherry, elderberry, and others [3,4,5]. Glucose is commonly attached to the aliphatic, cyclic, aromatic, and heterocyclic CNG aglycones [4]. Overconsumption of plant CNGs in humans and animals can lead to acute toxicity, which arises due to the production of hydrogen cyanide from the CNGs [6]. The rate of hydrogen cyanide production from CNGs is dependent on both endogenous and gut bacterial β-glycosidase activity [6]. Hence, measurement of CNGs in plants and foods is critical to understanding phytochemical defenses and possible health risks.

Elderberry (*Sambucus nigra* L.) belongs to the Adoxaceae family and genus Sambucus. *Sambucus nigra* L. is one of the most common species in this genus and is native to most of Europe [7]. Elderberry is emerging as a functional food due to its richness in many bio-active compounds including (poly)phenolics and terpenoids [8]. In addition to the above health-promoting compounds, elderberry also accumulates toxic CNGs in different tissues, such as stems, leaves, flowers, seeds, and berries [9,10]. Several species of Sambucus, including American elderberry (*Sambucus nigra* L. subsp. *canadensis* (L.) Bolli), have been reported to specifically contain the cyanogenic glycosides amygdalin, dhurrin, linamarin, (*R*)-prunasin and (*S*)-prunasin (sambunigrin) (Figure 1) [8]. The levels of CNGs in different parts of elderberry vary widely and were found to vary further during food processing. It has been found that total CNGs were higher in stems and green berries than in juice and seeds [10]. CNG content is considerably reduced (by 44–96%) in processed elderberry juice, tea, liqueur, and spread [11]. 

To date, high-performance liquid chromatography (HPLC), gas chromatography-mass spectrometry (GC–MS), and liquid chromatography coupled to tandem mass spectrometry (LC–MS/MS) have been used for quantitative analysis of CNGs in plant samples [12,13,14]. Among these, LC–MS/MS operated in the multiple reaction monitoring (MRM) mode is a highly sensitive and selective technique for quantitative analyses [15]. LC–MS/MS MRM methods isolate precursor ions, subsequently fragment them, and monitor the product ions. It increases the signal-to-noise ratio and, hence, the sensitivity of the quantification. For LC–MS/MS analyses of CNGs, electrospray ionization (ESI) has been preferably used in the positive ion mode [10,14,16]. It has been found that CNGs usually form sodium adducts [M + Na]^+^ instead of [M + H]^+^ molecular ions. Because of this, [M + Na]^+^ adducts of CNGs are typically selected as precursor ions in published MRM-based methods [14]. However, it has been reported that most [M + Na]^+^ adducts of CNGs, i.e., dhurrin, linamarin, (*R*)-prunasin, and (*S*)-prunasin (sambunigrin), do not generate product ions needed for LC–MS/MS MRM analyses, presumably due to the high stability of their sodium adducts [10]. The formation of highly stable [M + Na]^+^ adducts makes the quantification of dhurrin, linamarin, and amygdalin using LC-MS MRM difficult, if not impossible. In addition, the formation of multiple adducts, such as [M + H]^+^, [M + Na]^+^, and [M + K]^+^ in LC–MS/MS may also reduce the sensitivity of the analyses. This likely results in inconsistent data as the equilibria and dynamics among the adducts can change with mobile phases or sample matrix. 

Here we report a novel UHPLC–MS/MS MRM-based method to quantify CNGs. The first objective of this study was to optimize the chromatographic conditions for the separation of (*R*)-prunasin and (*S*)-prunasin (sambunigrin). Different columns, mobile phase gradients, and additives were investigated. We first investigated the impact of mobile-phase additives (i.e., formic acid and ammonium formate) on the ionization, separation, and sensitivity of major CNGs reported in the elderberry, such as dhurrin, linamarin, amygdalin, (*R*)-prunasin, and sambunigrin. Ammonium formate was chosen as the preferred mobile phase additive because it formed [M + NH_4_]^+^ adducts that fragmented more easily to produce product ions suitable for MRM analyses. Using the optimal mobile phase additive, we further developed and validated a UHPLC–MS/MS MRM for accurate quantification of CNGs in elderberry samples.

## 2. Materials and Methods

### 2.1. Chemicals and Reagents

Linamarin, dhurrin, amygdalin, (*R*)-prunasin, ammonium formate, and umbelliferone were purchased from Sigma-Aldrich (St. Louis, MO, USA). (*S*)-Prunasin (sambunigrin) was procured from Santa Cruz Biotechnology, Inc., (Dallas, TX, USA). Optima LC/MS-grade acetonitrile (ACN), methanol, and formic acid were obtained from Fisher Chemical (Hampton, NH, USA). Milli-Q, 18 MΩ water was for UHPLC-MS and was obtained from an in-house Millipore water purifier (Burlington, MA, USA).

### 2.2. North American Elderberry Samples 

The leaf and fruit samples of “Ozark” genotype of American elderberry utilized for this study were collected from the Horticulture and Agroforestry Research Farm (New Franklin, MO, USA) in the 2022 growing season (May–October). Samples were immediately frozen at collection, freeze-dried using a Labconco FreeZone freeze dryer (Kansas City, MO, USA), and ground into a fine powder. These samples were stored in Falcon tubes at −20 °C until further processing. 

### 2.3. Sample Preparation 

Fruit and leaf samples (10 ± 0.06 mg) were accurately weighed into glass vials. Extractions were carried out by adding 1.0 mL of 80% methanol/20% water containing the internal standard umbelliferone (18 µg mL^−1^). All samples were sonicated and then agitated for 18 h at room temperature. Afterwards, the samples were centrifuged at 3500 rpm for 20 min. The supernatants (~900 µL) were transferred into Eppendorf tubes and centrifuged at 15,000 rpm for 15 min. The supernatants were transferred into glass vials and dried using a Labconco RapidVac evaporator (Kansas City, MO, USA). Extracts were reconstituted in methanol before solid-phase extraction (SPE) and LC–MS/MS analysis. SPE purifications were carried out using Oasis® HLB 1cc (30-mg) extraction cartridges per the procedure described by Appenteng and co-authors, with some modifications [10]. Briefly, an SPE cartridge was conditioned with 2 mL of methanol and equilibrated with 2 mL of water. The sample was loaded onto the column, followed by washing with 0.1% formic acid in water. Finally, CNGs were eluted with 2 mL of 0.1% formic acid in methanol. The extracts were dried under nitrogen gas for further reconstitution.

### 2.4. UHPLC-MS/MS Analysis 

The chromatographic (mobile phase, gradient, column, and injection volume) and multiple reaction monitoring mode (MRM) conditions for quantitative measurements of CNGs were optimized using a Waters ACQUITY UPLC system coupled with a Waters Xevo TQ-XS triple quadrupole mass spectrometer (UHPLC-MS/MS, Milford, MA, USA). Chromatographic separations were performed using an ACQUITY UPLC HSS T3 column (1.8 µM, 2.1 mm × 100 mm, Milford, MA, USA) and a flow rate of 0.5 mL min^−1^. Mobile phases A and B were 2 mM ammonium formate in water and 100% ACN, respectively. The column temperature was set at 45 °C and the injection volume was 2 µL. The analysis time was 18 min and the elution gradient for mobile B was as follows: initially equilibrated at 1% B; 0–12 min, 1–12% B; 12–12.5 min, 12–95% B; 12.5–14.5 min, 95% B; 14.5–15 min, 95–5% B; 15–18 min, 5–1% B; 18 min. MassLynx software (version 4.1, Waters, Milford, MA, USA) was used for instrument control and data processing. The data related to quantification was processed using Waters QuanLynx software Ver 4.2 (Milford, MA, USA).

Mass spectra were acquired in positive electrospray ionization (+ESI) mode. Detection of all analytes was performed using MRM mode. The source temperature and desolvation temperature were maintained at 150 °C and 500 °C, respectively. Nitrogen and argon were used as the desolvation gas (1000 L h^−1^) and collision gas (0.14 mL min^−1^), respectively. 

### 2.5. Method Validation

The optimized method was validated for linearity, limit of quantification (LOQ), limit of detection (LOD), carry-over, repeatability, accuracy (percent recovery), and matrix effects, according to the US Food and Drug Administration (FDA) bioanalytical validation guidelines [17]. Linearity for each analyte was assessed by preparing ten-point solvent-based calibration curves in concentrations ranging from 1 to 500 ng mL^−1^. For each analyte, LOD and LOQ values were calculated using the following formula: LOD = 3.3 σ/S and LOQ = 10 σ/S. Where, σ and S were the standard deviation of the analytical background response and the slope of the calibration curve, respectively [18]. Sample carry-over was assessed by injecting five blank samples following the highest calibration standard concentration of each analyte. Repeatability and accuracy (percent recovery) for each analyte were assessed at mid-level concentration (32 ng mL^−1^) by injecting three repeated injections at three levels in a single sequence. For estimation of the matrix effect (%ME), a standard mixture solution (100 ng mL^−1^) was spiked into pooled fruit and leaf matrices and calculated as follows: %ME = [((As − Ans)/Abs) − 1] × 100, where As, Ans, and Abs represent the area of analytes in a spiked sample, a non-spiked sample, and a standard solution spiked in the solvent, respectively. 

### 2.6. Statistical Analysis

Results were expressed as a mean ± standard deviation (SD) of three biological replicates. Statistical analyses were performed using Microsoft Excel (version Microsoft 365).

## 3. Results

### 3.1. Optimization of Chromatographic Conditions

#### 3.1.1. Impact of the Gradient Elution Program and Injection Volume on the Separation of Prunasin Isomers

We first applied the reported chromatographic conditions [10] to the separation of CNGs and, indeed, found that (*R*)-prunasin and (*S*)-prunasin (sambunigrin) coeluted (Appendix A). Hence, we tested several different gradient elution programs and two different columns. The optimal gradient elution program for the separation of all CNGs, including (*R*)-prunasin and (*S*)-prunasin (sambunigrin) isomers is included in the Materials and Methods section. Briefly, the gradient started at 99% mobile phase A (0.1% formic acid in water) and 1% mobile phase B (acetonitrile-containing additive). Under this optimal gradient, all CNGs eluted before 12 min (Figure 2). Both (*R*)-prunasin and (*S*)-prunasin (sambunigrin) were separated satisfactorily, with a chromatographic separation resolution R = 1.3, nearly achieving baseline separation (Figure 3). Results of an injection volume optimization study showed that among the studied injection volumes (2, 3, and 5 µL), the 2 µL injection volume yielded the best separation resolution for (*R*)-prunasin and (*S*)-prunasin (sambunigrin) isomers (Figure 4). Injection volumes higher than 2 µL caused peak fronting. Larger injection volumes also resulted in significant overlap (3 µL) or complete overlap (5 µL) of *(R)*-prunasin and (*S*)-prunasin (sambunigrin). The effect of the column on peak shape and resolution of CNGs was also tested using ACQUITY UPLC HSS T3 C18 (1.8 µM, 2.1 mm × 100 mm) and ACQUITY Premier BEH C18 (1.7 µm, 2.1 mm × 100 mm) columns. In both cases, the peak resolutions were the same. However, peak splitting, mainly of linamarin was observed, when ACQUITY Premier BEH C18 was used (Appendix A). Hence, we decided to continue to use the ACQUITY UPLC HSS T3 C18 column for this study.

#### 3.1.2. Effect of Mobile Phase Additives on MRM Transitions

In this study, Waters IntelliStart ^®^ application software (ver 4.2) was used to generate product ions for all [M + Na]^+^ ions of CNGs (Appendix A). However, for most of the CNGs, the precursor ion and their corresponding product ions failed to produce a peak in the MRM chromatogram (Appendix A), except for (*R*)-prunasin and (*S*)-prunasin (sambunigrin). Hence, we studied the effect of ammonium formate as a mobile phase additive on improving MRM fragmentation and analyses. MRM parameters such as key transitions, cone voltage, and collision energy (as shown in Table 1) were obtained for [M + NH_4_]^+^ adducts of CNGs using Waters IntelliStart ^®^ application and were used to generate the MRM chromatograms. The MS2 fragmentation pattern of [M + NH_4_]^+^ adducts of all CNGs studied is shown in Appendix A. 

We further assessed the impact of the addition of 0.1% formic acid in an aqueous phase containing 2 mM ammonium formate. Here, we found that the addition of 0.1% formic acid in the aqueous phase containing 2 mM ammonium formate decreased the signal intensities of CNGs in the MRM chromatogram (Figure 2D). We also tested the effects of different concentrations of ammonium formate (2–10 mM)-containing aqueous phases on the signal intensities of the CNGs, including isomer separation. In this study, signal intensities of CNGs were found to decrease in MRM chromatograms with increasing concentrations of ammonium formate in the aqueous phase (Figure 2).

### 3.2. Optimization of Mass Spectrometric Conditions

The aim of this study was to achieve better sensitivity and selectivity; therefore, we decided to develop an MRM-based LC–MS/MS method for the simultaneous quantification of CNGs, including (*R*)-prunasin and (*S*)-prunasin (sambunigrin) isomers. The details of the optimized MRM parameters for each CNG are summarized in Table 1. MRM parameters such as precursor and product ions, cone voltage, and collision energy for each CNG were auto-optimized and obtained by using the IntelliStart^®^ autotune application, which is a feature of Waters MassLynx software (ver 4.2). These MRM parameters were initially tested by generating the MRM chromatograms. The resultant MRM chromatograms were acceptable, showing clear and distinct peaks for each CNG.

### 3.3. Method Validation

The results of our method validation studies are summarized in Table 2. Calibration curves for all CNGs studied showed good linearity within the concentration range studied (1–500 ng mL^−1^) and with squared correlation coefficients (R^2^) ranging from 0.995 to 0.999. The LOD and LOQ values indicate the lowest concentration of analyte that can be reliably detected and quantified, respectively [19]. The calculated LOD and LOQ values for each CNG are shown in Table 2. The results of intra-assay precision or repeatability were expressed in terms of the relative standard deviation (RSD). The observed RSD values were within the acceptable limit (<15%), as per Food and Drug Administration guidelines [17]. Similarly, the accuracy of the results, expressed in terms of the percent recovery (%RE), was within an acceptable range (80–120%) (Table 2). 

Matrix effects (%ME) were assessed in elderberry leaf and fruit tissues in terms of signal suppression or enhancement due to co-elution of matrix components. Before SPE sample clean-up, the observed matrix effect for the leaf and fruit tissue samples ranged between −23.6 to 45.6% (leaf) and 1.2 to 14.9% (fruit). A positive value indicates signal enhancement, and a negative value denotes ion suppression. If signal suppression or enhancement is greater than 20%, then the matrix effect should be addressed according to the SANTE guidelines [20]. Our results showed that the matrix effects observed in the fruit tissue samples were acceptable. However, ion enhancement occurred in the leaf tissue matrix. Hence, we performed a sample clean-up by SPE which decreased the matrix effects, especially in leaf samples (Appendix A).

### 3.4. (R)-prunasin and Sambunigrin in Elderberry Fruit and Leaf Samples

We measured CNGs, including (*R*)-prunasin and (*S*)-prunasin (sambunigrin) in fruit and leaf samples of the American elderberry varietal “Ozark”, and the results are shown in Figure 5. Of the CNGs studied, (*R*)-prunasin was only detected in leaf samples. In fruit samples, we did not detect any CNGs. 

## 4. Discussion

Due to the potential toxicity of CNGs, quantifying trace levels of CNGs in elderberry is essential to ensure its food safety. Because of this, successful chromatographic separation of CNGs is critical for their accurate quantification, especially for the two isomers (*R*)-prunasin and (*S*)-prunasin (sambunigrin). These prunasin isomers have the same molecular weight and produce the same product ions (Appendix A). In addition, a previous study reported that these isomers were not uniquely separated by UHPLC using a C18 column, making it impossible to differentiate one from another [10]. In this study, we first aimed to separate them chromatographically. Initially, chromatographic conditions, including column, gradient, and mobile phase additives, were optimized to separate CNGs, particularly (*R*)-prunasin and (*S*)-prunasin(sambunigrin) isomers. In preliminary experiments, we found that sample injection volume had a considerable impact on the (*R*)-prunasin and (*S*)-prunasin (sambunigrin) peak shape and resolution. Generally, an injection solvent with a high elution strength can cause peak distortion. Therefore, it is recommended to either use a low injection volume or reconstitute the final sample in a more aqueous solvent. In our case, we had reconstituted our sample in methanol. Hence, the injection volume was also optimized, and this study showed that 2 µL injection volumes yielded better separation of (*R*)-prunasin and (*S*)-prunasin (sambunigrin) isomers. The effect of the use of two different columns on chromatographic separation of CNGs was also tested. In both cases, the peak resolutions were the same. However, the linamarin peak split when the BEH C18 column was used. This could be due to the higher polar nature of linamarin. Compared to other CNGs, linamarin lacks the nonpolar benzene moiety, making it the most polar compound of the five CNGs and less amenable to BEH C18 column separation. On the contrary, the HSS T3 C18 column chemistry is designed to retain both polar and nonpolar compounds, and it was found to be more suitable for the separation of CNGs that have the polar carbohydrate moiety.

It has been previously shown that CNGs pose significant challenges not only in chromatographic separation but also in MRM quantitative analyses because most of the [M + Na]^+^ ions from CNGs do not generate fragment ions suitable for MRM analyses [10]. As observed in the previous study, we also found that all CNGs typically form sodium [M + Na]^+^ instead of [M + H]^+^ ions in positive ESI mode, even when 0.1% formic acid was used as an aqueous phase [10,14]. The source of sodium adduct formation likely originates from the source material used to isolate the standard and the high affinity of CNGs for Na^+^. Lesser amounts could also arise from solvent impurities, glassware, or related sample handling [21], but we make efforts to minimize these in our lab. Waters IntelliStart ^®^ application software(ver 4.2) autogenerated molecular and predicted product ions of [M + Na]^+^ adducts of CNGs; however, the physical experiments failed to generate peaks during the MRM analysis for linamarin, dhurrin, and amygdalin. This indicates the high stability of the [M + Na]^+^ adducts of linamarin, dhurrin, and amygdalin. High stability of parent [M + Na]^+^ adduct ions results in low intensity of product ions, as reported previously [22]. In the previous study, the addition of ammonium salt additives in the aqueous phase was recommended to improve chromatographic performance and LC–MS/MS analysis, as [M + NH_4_]^+^ adducts are easy to fragment compared with [M + Na]^+^ adducts [23]. Similarly, we found that the addition of ammonium formate (2 mM) to the aqueous phase facilitates the formation of ammonium [M + NH_4_]^+^ adducts of all CNGs studied. Our results also showed that all CNGs produced distinct product ion peaks in the MRM chromatograms. This finding confirmed that [M + NH_4_]^+^ adducts of all CNGs studied can be easily fragmented, and the characteristic fragments of each CNG can be used to develop the MRM method.

Acidification of the mobile phase, such as adding 0.1% formic acid to an aqueous phase containing ammonium formate, has been reported to further increase the signal intensity of analytes [24,25]. Hence, we assessed the impact of the addition of 0.1% formic acid in an aqueous phase containing 2 mM ammonium formate. However, the results of our study showed the addition of 0.1% formic acid in the aqueous phase containing 2 mM ammonium formate actually suppressed signal intensities of CNGs in the MRM chromatograms. Similarly, we found that a higher concentration of ammonium formate (>2 mM) in the aqueous phase also suppressed signal intensities of CNGs in the MRM chromatogram. Altogether, results showed that optimized chromatographic and mass spectrometric conditions were suitable for UHPLC–MS/MS MRM accurate quantification of CNGs reported in the elderberry. 

Method validation is crucial to ensure the developed method is reliable, reproducible, and fit for purpose [26]. Hence, the optimized method was validated for linearity, LOD, LOQ, carry-over, repeatability, accuracy, and the matrix effect. These results altogether confirmed that the developed method was precise and accurate. No observed carryover was found. The matrix effects observed in the fruit tissue samples were acceptable. However, sample clean-up by SPE is essential for the leaf tissue, as ion enhancement occurred. 

Both (*R*)-prunasin and (*S*)-prunasin (sambunigrin) have been reported in elderberry species [8]. In a previous study, the separation of (*R*)-prunasin and (*S*)-prunasin (sambunigrin) was reported to be difficult using reversed-phase UHPLC and a C18 column [10]. However, (*R*)-prunasin and (*S*)-prunasin (sambunigrin) were successfully separated and quantified in elderberry leaves and flowers using HPLC with diode array detection (DAD) in a different study [27]. However, the sensitivity and specificity of HPLC–DAD are lower when compared to UPLC–MS/MS. We used the optimized and validated UHPLC–MS/MS MRM method to quantify CNGs in fruit and leaf samples of the American elderberry. Interestingly, we did not find (*S*)-prunasin (sambunigrin) in fruit and leaf samples, a characteristic CNG in the elderberry species studied. This fact suggests that the characteristic presence of elderberry glycosides reported in the literature in elderberry species needs to be verified in larger experiments involving various elderberry species and their individual parts. 

## 5. Conclusions

In this study, we report an optimized UHPLC–MS/MS MRM method for rapid analysis of elderberry CNGs. We uniquely demonstrate the importance of ammonium formate as an aqueous phase additive to improve the chromatographic performance of CNGs in MRM-based UHPLC–MS/MS analyses. Additionally, we showed that the isomers (*R*)-prunasin and (*S*)-prunasin (sambunigrin) can be separated via UHPLC using a C18 column, and reliably quantified in plant samples. The MRM-based UHPLC–MS/MS method presented in this study is novel, sensitive, and reliable enough for the simultaneous quantification of CNGs, including (*R*)-prunasin and (*S*)-prunasin (sambunigrin), in elderberry samples. Of the CNGs studied, *(R)*-prunasin was only detected in leaf samples. Interestingly, we did not detect (*S*)-prunasin (sambunigrin), a characteristic CNG in the Sambucus species studied, indicating the need for further research on its presence and distribution in other Sambucus species.

## Figures and Tables

**Figure 1 metabolites-14-00360-f001:**
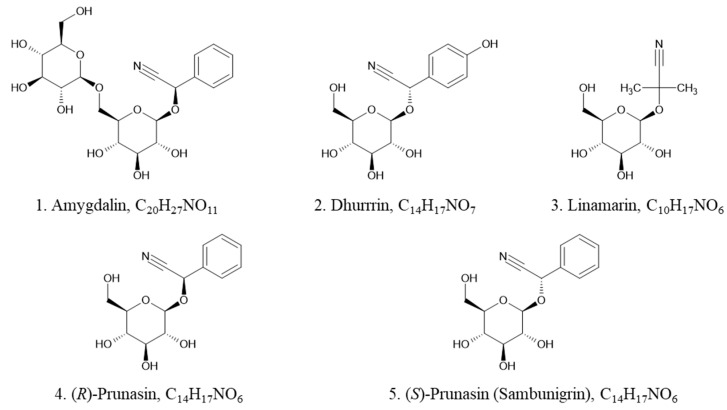
Structures of five common CNGs in elderberry: 1. amygdalin, 2. dhurrin, 3. linamarin, 4. *(R)*-prunasin, and 5. *(S)*-prunasin (sambunigrin).

**Figure 2 metabolites-14-00360-f002:**
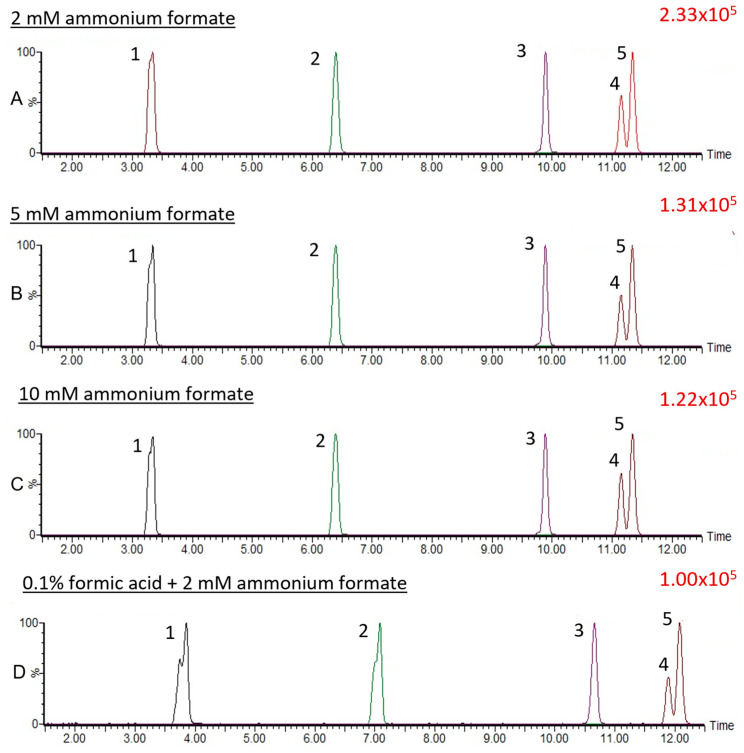
Effect of different concentrations of ammonium formate (2–10 mM) as a mobile phase additive on the separation of the 5 CNGs: (**A**) 2mM ammonium formate, (**B**) 5 mM ammonium formate, (**C**) 10 mM ammonium formate, (**D**) 0.1% formic acid + 2 mM ammonium formate: 1. linamarin, 2. dhurrin, 3. amygdalin, 4. (*R*)-prunasin, and 5. (*S*)-prunasin (sambunigrin) in ESI(+).

**Figure 3 metabolites-14-00360-f003:**
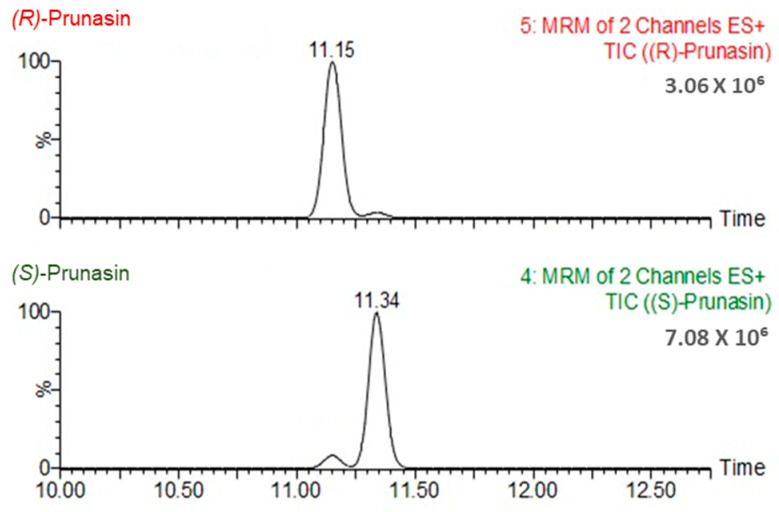
Multiple reaction monitoring (MRM) chromatograms of isomeric separation of (*R*)-prunasin and (*S*)-prunasin (sambunigrin).

**Figure 4 metabolites-14-00360-f004:**
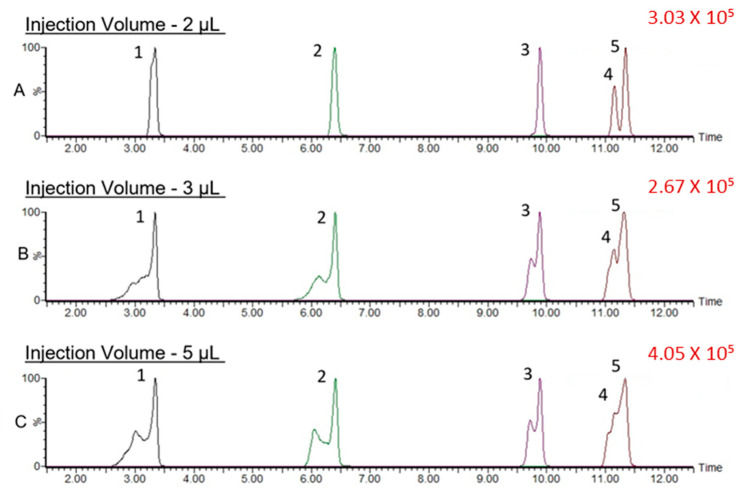
Effect of different injection volumes (2, 3, and 5 µL) on the chromatographic separation and peak shape of cyanogenic glucoside standards: 1. linamarin, 2. dhurrin, 3. amygdalin, 4. (*R*)-prunasin, and 5. (*S*)-prunasin (sambunigrin). (**A**): Resultant chromatogram using an injection volume of 2 µL. (**B**): Resultant chromatogram using an injection volume of 3 µL. (**C**): Resultant chromatogram using an injection volume of 5 µL.

**Figure 5 metabolites-14-00360-f005:**
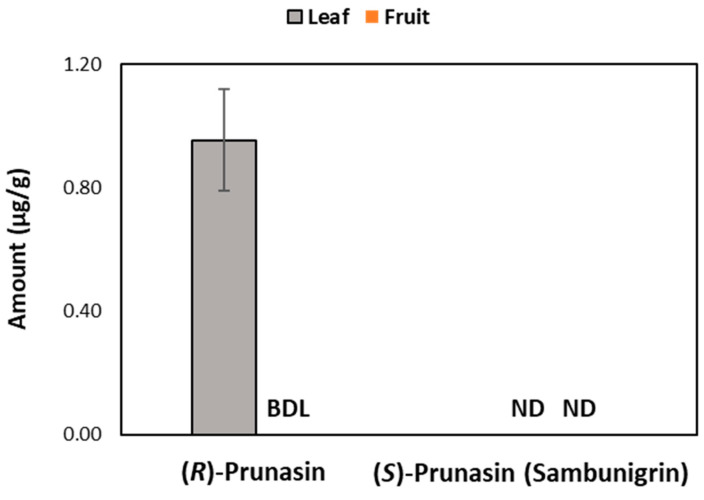
Levels of (*R*)-prunasin and (*S*)-prunasin (sambunigrin) in dried leaf and fruit samples of Ozark elderberry samples measured by UHPLC-MS/MS MRM. BDL = below detection limit and ND = not detected.

**Table 1 metabolites-14-00360-t001:** Optimized multiple-reaction monitoring mode (MRM) conditions for the analysis of elderberry cyanogenic glycosides using ESI(+) mode.

Sr. No.	Compound	Rt (min)	Ion type	Transition (*m*/*z*)	Cone (V)	Collision (eV)
1.	Linamarin	3.34	Quantifier	265.40 > 180.01	4	8
			Qualifier	265.40 > 162.92	4	10
2.	Dhurrin	6.40	Quantifier	329.45 > 131.90	8	10
			Qualifier	329.45 > 84.80	8	26
3.	Amygdalin	9.89	Quantifier	475.46 > 84.87	24	30
			Qualifier	475.46 > 144.92	24	20
4.	(*R*)-Prunasin	11.14	Quantifier	313.45 > 180.01	20	8
			Qualifier	313.45 > 144.92	20	12
5.	(*S*)-Prunasin (Sambunigrin)	11.34	Quantifier	313.45 > 180.01	20	8
			Qualifier	313.45 > 144.92	20	12

**Table 2 metabolites-14-00360-t002:** Results of linearity determined for the range of 1–500 ng mL^−1^, limit of detection (LOD), limit of quantification (LOQ), repeatability, accuracy, and matrix effect (%ME) of the established method for cyanogenic glycosides. RSD, relative standard deviation, and %RE, percent recovery.

Sr. No.	Compound	R^2^	LOD (ng mL^−1^)	LOQ (ng mL^−1^)	Repeatability (RSD)	Accuracy (%RE)	Matrix Effect (%ME)	
							Leaf Tissue	Fruit Tissue
1.	Linamarin	0.999	0.0012	0.0037	8.35	93.9	−23.6	1.2
2.	Dhurrin	0.999	0.0012	0.0035	7.53	96.4	45.6	7.3
3.	Amygdalin	0.998	0.0008	0.0023	4.81	86.4	32.4	9.2
4.	*(R)*-Prunasin	0.997	0.0020	0.0061	5.37	100.4	8.3	13.1
5.	(*S*)-Prunasin (Sambunigrin)	0.995	0.0009	0.0027	6.52	87.6	31.4	14.9

## Data Availability

The data presented in this study are available on request from the corresponding author.

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
