# Peer review of "A Novel UHPLC-MS/MS Based Method for Isomeric Separation and Quantitative Determination of Cyanogenic Glycosides in American Elderberry"

_metabolites, 2024, doi:10.3390/metabo14070360_

Round 1
Reviewer 1 Report
Comments and Suggestions for Authors
In this submitted paper, the authors tried to establish a novel Multiple Reaction Monitoring (MRM)-based ultra-high performance liquid chromatography-tandem mass spectrometry (UHPLC-MS/MS) method for simultaneous separation and quantification of five cyanogenic glycosides (CNGs) such as amygdalin, dhurrin, linamarin, (R)-prunasin and (S)-prunasin. Overall, the design of the research is reasonable; the topic is interesting since it addresses the challenging issue with regard to the separation and quantification of CNGs using the LC-MS/MS method. In addition, the writing language is acceptable along with the comprehensive discussion provided over the outcomes achieved. Considering such key points, this paper is recommended to be accepted for publication in the Metabolites without any further revision.
Author Response
We are thankful to the reviewer for providing valuable insights into the overview of our manuscript. We appreciate the reviewer's positive comments and the recommendation of our manuscript for publication.

Reviewer 2 Report
Comments and Suggestions for Authors
The manuscript describes in a systematic way the optimization and validation of a mass spectrometry-based method for the reliable quantification of cyanogenic glycosides in elderberry plant. The application of the method interestingly showed the presence of R-prunasin rather than S-prunasin in the studied elderberry species, and that it contrary to what would be expected, highlighting the importance of the work.
The manuscript is clear and well written. For future work, it would have been better to compare an HSS rather than a BEH column to the HSS T3 column, to rule out the added effect of having a different stationary phase support that might result in added interactions with the compounds. Other than that I only have minor comments to help improve the manuscript:
-R and S denoting the prunasin isomers should be italicized.
-Page 6, lines 206 and 209: it should be signal intensities not single intensities.
-Page 9, line 288: it should be sodium adduct.
Comments on the Quality of English LanguageThe English language of the manuscript is of a good quality. Some minor typos/grammatical mistakes were observed, but nothing critical. A good revision before the final publication should resolve this issue.
Author Response
We are grateful to the reviewer for offering valuable insights into the overview of our manuscript. We appreciate the reviewer's positive comments. Definitely, we will consider your suggestion regarding the use of HSS columns for a comparative study in future research.
- -Rand S denoting the prunasin isomers should be italicized.
Response: Corrected as suggested.
- -Page 6, lines 206 and 209: it should be signal intensities not single intensities.
Response: Corrections have been made as per suggestion.
- -Page 9, line 288: it should be sodium adduct.
Response: Corrected as suggested.

Reviewer 3 Report
Comments and Suggestions for Authors
- The authors demonstrated separation using different concentrations of ammonium format. Were other mobile phase additives, such as ammonium acetate, ammonium fluoride, or just formic acid, tested to achieve separation?
- Typically, in reverse-phase gradient elution, the final reconstitution is done in a more aqueous solvent (5% or 10% organic solvent). line 112, can the authors comment on why reconstitution was done in methanol? Could this be the reason for the poor peak shape observed with higher injection volumes?
- line 113, please include a brief description of the SPE protocol in the main text.
- In Section 3.1.2, the authors mention that the fragmentation of M+Na adducts did not show peaks, while Table S1 depicts transitions for the M+Na adduct. It is unclear which adduct is used in the study. Please clarify this section.
- Were labeled internal standards used to investigate the matrix effect or quantification from the biological matrix?
- Line 247 states that the matrix effect should not exceed 20%. However, Table 2 shows values of 32.4% and 45.6%. Please explain this discrepancy.
Author Response
The authors express gratitude to the editor and the reviewer for reviewing our manuscript and offering us the opportunity to enhance its quality. We have thoroughly considered the comments and revised the manuscript accordingly, using track changes and highlighting the modifications in red. Our responses to your comments are outlined below.
Reviewer #3:
- The authors demonstrated separation using different concentrations of ammonium format. Were other mobile phase additives, such as ammonium acetate, ammonium fluoride, or just formic acid, tested to achieve separation?
Response: We attempted to use 0.1% formic acid, but we were unable to obtain MRM chromatograms for linamarin, dhurrin, and amygdalin. We did not experiment with ammonium acetate and ammonium fluoride. However, it is a good suggestion, and we will consider using these mobile phase additives in our future studies.
- Typically, in reverse-phase gradient elution, the final reconstitution is done in a more aqueous solvent (5% or 10% organic solvent). line 112, can the authors comment on why reconstitution was done in methanol? Could this be the reason for the poor peak shape observed with higher injection volumes?
Response: Thank you for your question, we appreciate your input. We agree that peak shapes can be better if ones dissolve the sample in a more aqueous solvent. However, the final percent organic depends on the solubility of your analyte, relative retention, and whether or not you are doing targeted or nontargeted types of analyses. Unfortunately, we did not test this in our study, but have added the following text to the discussion [Line-272-275: “Generally, an injection solvent with high elution strength can cause peak distortion. Therefore, it is recommended to either use a low injection volume or reconstitute the final sample in a more aqueous solvent. In our case, we had reconstituted our sample in methanol”.
- line 113, please include a brief description of the SPE protocol in the main text.
Response: We have added a brief description of the SPE protocol in the main text.
- In Section 3.1.2, the authors mention that the fragmentation of M+Na adducts did not show peaks, while Table S1 depicts transitions for the M+Na adduct. It is unclear which adduct is used in the study. Please clarify this section.
Response: The Waters IntelliStart ® application software generated product ions for all [M + Na]+ ions of CNGs (Table S1). However, for most of the CNGs, the precursor ion and their corresponding product ions failed to produce a peak in the MRM chromatogram (Figure S3), except for (R)-prunasin and (S)-prunasin (sambunigrin). We hope this clarifies the issue. (Similar changes have been made in the manuscript as well; lines 200-202).
- Were labeled internal standards used to investigate the matrix effect or quantification from the biological matrix?
Response: No, we have not used a labeled internal standard.
- Line 247 states that the matrix effect should not exceed 20%. However, Table 2 shows values of 32.4% and 45.6%. Please explain this discrepancy.
Response: Table 3 shows the observed matrix effect for the leaf and fruit tissue samples before SPE sample clean-up. We have corrected the sentence to reduce ambiguity.
- The english is readable and generally clear. However, several syntax errors are present.
Response: As per suggestion, we have tried to correct syntax and grammatical errors.
